# Cognitive Ability, Personality, and Psychopathology: A Stormy Relationship

**DOI:** 10.3390/jintelligence12100096

**Published:** 2024-09-29

**Authors:** Roberto Colom, Pei-Chun Shih Ma

**Affiliations:** Department of Biological and Health Psychology, Universidad Autonoma de Madrid, 28049 Madrid, Spain; pei_chun.shih@uam.es

**Keywords:** intelligence, cognitive ability, personality, psychopathology

## Abstract

Cognitive and non-cognitive traits are frequently analyzed in isolation. However, there is an increasing acknowledgment that their interplay should be considered for enhancing our understanding of human psychological differences. Testing both traits in the same sample of individuals is desirable when addressing their relationships. Here, for that purpose, 299 university students from Spain (mean age = 18.5 years., 83% female) completed a cognitive ability battery comprised by nine tests, the NEO-FFI for assessing the big five personality traits, and the SCL-90-R for evaluating a range of subjective psychopathological symptoms. This resulted in 23 cognitive and non-cognitive variables that were submitted to a data reduction providing four factors: (1) neuroticism/p, (2) cognitive ability/*g*, (3) agreeableness/A, and (4) introversion/I. Summary factor scores revealed a positive correlation between p and I (0.47), along with negative correlations of A with p (−0.26) and with *g* (−0.24), and a negative correlation between A and I (−0.16). These factors were related to some degree even when the assessment of the cognitive and non-cognitive variables cannot be considered straightforwardly comparable because the former was performance based, whereas the later was based on self-reports. Conceptual and methodological implications are discussed regarding the three-way relationship among cognitive ability, personality, and subjective psychopathological symptoms.

## 1. Introduction

Scientific research has shown that cognitive and non-cognitive psychological dimensions do interact. Indeed, there is a long tradition of theoretical models considering their mutual relationships ([2]; [28]; [31]). However, in the special issue published in the *Journal of Intelligence* addressing, once again, the cognitive ability–personality integration, [9] ([9]) wrote “we are afraid that researchers will come back to their usual practice of focusing on a limited set of psychological constructs analyzed in isolation”. Their relationship seems stormy.

A large-scale meta-analysis considered the relationship between personality and cognitive ability (aka, intelligence) ([3]). Their key finding revealed that the relation between personality (self-reported) and cognitive ability (objectively measured) was moderate but meaningful from a theoretical standpoint. These researchers discussed “the intelligence-as-cause perspective” meaning that cognitive ability might influence differential personality configurations. Still, another interesting result was how personality facets relate to intelligence better (*adjusted multiple R* = 0.40) than general personality traits do (*adjusted multiple R* = 0.22).

[32] ([32]) meta-analyzed the relationships among 176 cognitive and non-cognitive variables from 1325 studies carried out in 50 countries finding a wide range of values. Thus, for instance, the corrected correlation between general cognitive ability (*g*) and the general factor of personality (P) was 0.36, between fluid ability (Gf) and anxiety, it was −0.28, between verbal ability (Gc) and suspiciousness, it was −0.15, and between visual processing (Gv) and depression, it was −0.32. They concluded that “the findings have implications for theories. For example, theories of self-regulation should further integrate personality traits and cognitive abilities”. We will revisit this integrative role of self-control later.

Regarding psychopathology, [1] ([1]) identified a ‘C factor’ representing cognitive disfunction and interpreted as a transdiagnostic dimension. Their systematic review of the evidence concluded that “compared with non-psychiatric controls, any form of psychopathology examined is associated with poorer performance across cognitive domains”. These researchers related their main finding (the presence of a C factor) to the p factor, representing generalized vulnerability to mental illnesses of heterogeneous seriousness, thus highlighting the relevance of cognitive deficits within the psychopathology realm. A related conclusion was achieved, for instance, by [6] ([6]) in their longitudinal follow-up (across four decades) of the Dunedin cohort. Higher p factor scores were related to lower neurocognitive functioning at age 3, greater cognitive decline from childhood to adulthood, and even older brain age.

[25] ([25]) analyzed the relationships among personality, personality disorders, and psychopathology, but they also related these three factors with everyday life maladaptivity using variables such as physical health, social support, loneliness, insomnia, and life satisfaction. Substantial overlap among personality, personality disorders, and psychopathology was found (*r* values between 0.70 and 0.92). Moreover, the general factor of personality and the factor summarizing personality disorders predicted maladaptivity (*r* values of −0.65 and 0.68, respectively). They concluded that “some persons do indeed have many desirable traits, whereas other persons have many undesirable traits”. General factors, according to their thoughts and to the obtained evidence, are not spurious.

This later conclusion was consistent with [18]’s ([18]) study and with [24]’s ([24]) evaluation of the available evidence regarding the debated general factor of personality (P). Judge et al. demonstrated that measures of self-esteem, locus of control, self-efficacy, and neuroticism tap the same higher-order latent factor. Once this common higher-order factor was controlled for, the specific factors were virtually meaningless. The common factor was mainly neuroticism but in a broad sense including two components: anxiety (dysphoria, stress) and self-evaluations (beliefs about one’s ability to control the environment). And this was Loehlin’s conclusion: “a general factor of personality is a fairly generalizable and quite readily measurable phenomenon (…) a failure of some particular scale of some particular inventory to load on an overall general factor of personality should be of no great concern” (p. 262). The jangle fallacy might be at play ([4]).

[22] ([22]) designed a comprehensive study for investigating the relationships among cognitive ability, personality, personality pathology, and psychopathology. The main aim was to test the likelihood of the so-called ‘big everything’. High-risk college students were assessed for that purpose and one of their key findings revealed (like in Oltmanns et al.’s study described above) remarkable overlap among personality, personality pathology, and psychopathology (*r* values between 0.71 and 0.87). However, cognitive ability was set apart (*r* values between 0.07 and 0.12) most probably because the measures tapping the cognitive factor were performance based instead of self-report based.

In summary, one reasonable conclusion, so far, is that it is complicated to find substantive relationships between cognitive and non-cognitive factors. This is complicated, but doable. Indeed, [3] ([3]) and [32] ([32]) meta-analytic findings underscored that there is a theoretically meaningful correlation between cognitive ability and personality. Furthermore, psychopathological vulnerability has something to do with cognitive ability ([1]; [6]). There are even research studies finding moderate relationships between p and *g* from early life. Thus, for instance, [16] ([16]) studied twins from 0.6 to 7.1 years of age and the correlation p × *g* ranged from −0.21 to −0.34, suggesting overlapping factors already present in the first years of life. Also, [30] ([30]) reported correlation values of 0.26 and 0.28 between a general factor of intelligence (*g*) and a general factor of personality (P). Therefore, these cognitive and non-cognitive factors seem moderately related, they are not independent.

Within the general research framework aimed at investigation of the simultaneous relationships among cognitive and non-cognitive factors, here we will measure cognitive abilities, basic personality traits, and subjective psychopathological symptoms. Studies considering these three factors are simultaneously rare. Cognitive ability will be measured through a battery comprising standardized tests taping reasoning or fluid ability, crystallized ability, visuospatial ability, and quantitative ability. Personality traits will be obtained from the NEO-FFI. Finally, subjective psychopathological symptoms will be measured by the (Symptom Checklist) SCL-90-R. The key prediction is that we will obtain moderate (medium) relationships among cognitive ability, personality, and psychopathology. For interpreting the correlation values (*r*), as estimates of effect size, we rely on [13]’s ([13]) recommendations: very small (0.05), small (0.10), medium/moderate (0.20), large (0.30), and very large (0.40 or greater) (see also [14]).

## 2. Method

### 2.1. Participants

Two hundred and ninety-nine first-year university psychology students from Madrid (Spain) took part in the present study. They gave informed written consent. Their mean age was 18.5 years (SD = 1.6) and 83% were female.

### 2.2. Measures

*Cognitive ability*. The cognitive battery was comprised of nine tests tapping abstract reasoning ability, verbal ability, visuospatial relations, and quantitative ability. The main features of the administered test are described next.

DAT-AR (abstract reasoning). This is a series test based on abstract figures. Fifty items are comprised by this test. Each item includes four figures following a given rule, and the participant must choose one of five possible alternatives properly completing the series. The score is the total number of correct responses. Completion time = 25 min.

RAPM (Raven Advanced Progressive Matrices). Each of the thirty-six RAPM items comprises a matrix figure with three rows and three columns. Among eight possible alternatives, the one completing the 3 × 3 matrix figure must be chosen (max. score = 36). Completion time = 40 min.

PMA-V (vocabulary). This is a synonym test that comprises 50 items. The meaning of four alternative words must be evaluated against a given word that serves as model. Only one alternative is correct. The score is the total number of correct responses. Completion time = 4 min.

Verbal comprehension. This test includes thirty items. Each item is based on the proper understanding of a given sentence that must be compared with three alternatives of whom only one carries the same thought. The score obtained is the total number of correct responses. Completion time = 15 min.

Verbal meanings. This test includes 20 items. Each item is composed of a sentence with an uppercase word. Participants must choose from among five words the one that keeps the proper meaning of the sentence. Items increase their complexity across the test. The score obtained is the total number of correct responses. Completion time = 4 min.

Rotation of solid figures. This test comprises 21 items. Each item includes a model figure, and five alternatives must be evaluated against it. The participant must evaluate which alternative can be rotated within a 3D space to fit the model figure. Only one alternative is correct. The score is the total number of correct responses. Completion time = 5 min.

Identical figures. This test includes 25 items. Each item comprises a model figure and five alternatives of whom only one is identical to the model figure. The figures increase their complexity across the test. The score obtained is the total number of correct responses. Completion time = 3 min.

Monedas-2 (quantitative reasoning). This is a quantitative reasoning test comprising 40 items. The items are based on the combination of the size of a series of coins (large, medium, and small), the digits put inside the coins to specify the number of them that participants must consider, and some numerical operations to make the necessary calculations to arrive at a given response (adding, subtracting, and so forth). Only one alternative is correct. The score obtained is the total number of correct responses. Completion time = 12 min.

PMA-N (rote calculation). This is a test that comprises 70 items. The participant must simply evaluate if a given sum is correctly or badly solved. For instance, 16 + 38 + 45 = 99? The score is the total number of correct responses. Completion time = 6 min.

*Personality Traits.* The basic personality traits of the five-factor model (extraversion, agreeableness, conscientiousness, neuroticism, and openness) were measured by the NEO-FFI, the 60 items version of the NEO-PI-R ([10]).

*Subjective Psychopathological Symptoms.* Psychopathological symptomatology was measured by the SCL-90-R ([11]; [15]). This test was designed to evaluate subjective experiences of psychopathology and taps nine dimensions: somatization (distress related to one’s body/physiological experiences), obsessive–compulsive (intrusive thoughts and compulsive actions), interpersonal sensitivity (self-perceived inadequacy/inferiority in relationships with others), depression (low mood and decreased sense of meaning), anxiety (anxious symptoms), hostility (aggressiveness), phobic anxiety (fears related to specific stimuli), paranoid ideation (persecutory cognitions), and psychoticism (psychotic behaviors).

### 2.3. Procedure

The cognitive ability tests (except the RAMP) were administered in two sessions in groups of forty students. In the first session, the following tests were administered: DAT-AR, PMA-V, and Monedas-2. In the second session, the following tests were administered: PMA-N, verbal meanings, identical figures, rotation of solid figures, and verbal comprehension. The RAPM, the NEO-FFI, and the SCL-90-R were completed online.

### 2.4. Analyses

The 9 cognitive ability variables were collapsed in 4 summary scores for obtaining estimates of reasoning/fluid ability (Gf = RAPM + DAT-AR), crystallized ability (Gc = PMA-V + verbal meanings + verbal comprehension), visuospatial ability (Gv = identical figures + solid figures), and quantitative ability (Gq = PMA-N + Monedas-2). Afterwards, a correlation matrix was computed for these 4 summary scores, the 5 personality traits measured by the NEO-FFI, and the 9 psychopathological symptom dimensions measured by the SCL-90-R. The complete correlation matrix, including means, standard deviations, asymmetry, skewness, and reliability values for the 23 variables, are shown in Appendix A. Finally, data reduction computations were conducted using exploratory principal axis factoring (PAF, eigenvalues greater than 1) followed by an oblique Promax rotation. Finally, factor scores were computed, and their correlation values were calculated. Additionally, and as suggested by the reviewers of the present report, we computed separate factor analyses for the cognitive ability variables, the personality traits, and the subjective psychopathological symptoms for documenting another approach to the dataset. The first principal unrotated factors for the three domains were considered for that purpose.

## 3. Results

Table 1 shows the correlation matrix comprising the four cognitive ability summary variables, the five personality traits, and the nine subjective psychopathological symptoms.

The correlation values for the cognitive ability variables ranged from 0.27 to 0.45, for the personality traits, they ranged from 0.00 to .28, and for the psychopathological dimensions, these values ranged from 0.41 to 0.77. Therefore, cognitive ability and psychopathology showed a noteworthy positive manifold, whereas the personality traits showed weaker (but not null) relationships among themselves.

Table 2 shows the factor matrix after computing the data reduction analysis.

F1 summarized the nine subjective psychopathological dimensions of the SCL-90-R along with the neuroticism (N) scale of the NEO-FFI. Therefore, this first factor seems to capture p (generalized vulnerability to psychopathology). F2 summarized the four cognitive variables (*g*). F3 was defined by the agreeableness (A) scale of the NEO-FFI along with the hostility and paranoid dimensions of the SCL-90-R. Finally, F4 was defined by the extraversion (E) scale from the NEO-FFI along with the interpersonal sensitivity dimension from the SCL-90-R. These four factors seem to organize the cognitive ability, personality, and psychopathological variables representing p, *g*, agreeableness, and introversion.

Next, four factor scores were computed (via the regression method) and their correlation values were obtained. The correlation between p and I was very large (*r* = 0.47, *p* < .01), between p and A was moderate (*r* = −0.26, *p* < .01), between *g* and A was also moderate (*r* = −0.24, *p* < .01), and between A and I was small (*r* = −0.16, *p* < .05). The correlation between *g* and p was small (*r* = 0.10, ns) as well as between *g* and introversion (*r* = 0.11, ns).

Appendix B shows the results for the separate factor analyses of the three domains, as detailed above. Here, cognitive ability was unrelated to the factors representing the personality traits as well as the subjective psychopathological symptoms. However, there was a large negative correlation (*r* = −0.60) between personality and psychopathology.

## 4. Discussion

### 4.1. Summary of the Main Findings

Here, we have shown that the cognitive and non-cognitive factors analyzed after the measurement of 23 psychological variables are related both within and between domains. These relationships, however, varied in their strength. Within domains, psychopathological symptoms and cognitive variables were more related than personality variables. Across domains, psychopathology (p) and introversion (I) showed a large positive correlation, whereas agreeableness (A) showed moderate correlations with both psychopathology (p) and cognition (*g*) as well as a small correlation with introversion (I). Therefore, the take home message might be that cognitive and non-cognitive factors are not independent, albeit their relationship is heterogeneous. Next, we discuss these findings step by step.

### 4.2. Cognition and Personality

[3]’s ([3]) meta-analysis reported a moderate correlation of 0.22 between intelligence/cognitive ability and personality. Here, we also found a moderate correlation of −0.24 between the identified agreeableness and cognitive factors. The negative sign of the correlation suggests that higher cognitive ability is moderately associated with lower agreeableness (A). Table 2 shows a positive factor loading (0.57) for the A NEO-FFI scale along with negative loadings for the hostility (−0.45) and paranoid (−0.38) dimensions of the SCL-90-R. Furthermore, the correlation between the cognitive and introversion factors was small and non-significant (0.11). Therefore, cognition is moderately related to some but not all personality traits.

### 4.3. Cognition and Psychopathology

[6] ([6]) and [5] ([5]) reported small correlation values, ranging from −0.11 to −0.17, between cognition and p (higher cognitive performance was therefore related to lower psychopathological vulnerability). Here, the correlation between cognition and p was 0.10 (ns). Therefore, cognitive differences are unrelated to greater or smaller values of psychopathological vulnerability in the sample of individuals analyzed here. One hypothesis considered by [5] ([5]) regarding the relationship between cognitive ability and p was that “deficits in intellectual functioning characterize p”. This seems in agreement with [1]’s ([1]) ‘C factor’ and may help to explain why we did not find statistically significant associations between these two factors here. University students can hardly be considered intellectually disabled or affected as a group by remarkable psychopathological symptoms.

### 4.4. Personality and Psychopathology

The correlation identified here between p and introversion was large, whereas the correlation between p and agreeableness was moderate. Table 2 shows a very strong p factor loaded by all the subjective symptom dimensions taped by the SCL-90-R along with the high loading of the NEO-FFI neuroticism (N) scale. We think this pattern makes sense. In fact, another hypothesis raised by [5] ([5]) was that “p represents a diffuse unpleasant affective state” (aka, neuroticism) (see [12]; [20]; [33]). It has been suggested that the N scales comprised in the usual personality questionnaires may tap some sort of unspecific risk for psychopathology ([26]). The distinction suggested by [18] ([18]), as discussed in the Introduction section, between the anxiety and the self-evaluative sides of N, may be relevant at this point of our discussion. The former may be more related to the non-cognitive domain of the human mind, whereas the later might be related to control processes associated with the intellect.

We also found here, however, that the extraversion, agreeableness, and conscientiousness scales of the NEO-FFI showed negative factor loadings on the p factor described in the previous paragraph, with values ranging from −0.20 to −0.31. This pattern may be consistent with the negative correlation (−0.70) reported by [25] ([25]) between p and P (the general factor of personality).

### 4.5. Three-Way Relationship

We finally turn to the simultaneous relationships among cognitive ability, personality traits, and subjective psychopathological symptoms. The study by [22] ([22]) described above supported the already discussed relationship between personality and psychopathology. However, they did not find any relationship between their cognitive factor and the personality or psychopathology factors. Nevertheless, they observed something that perhaps deserves attention: the loadings on the big everything factor of all the objective performance-based cognitive variables were zero, but the loading of their cognitive test based on the self-report format (Cognitive Failures Questionnaire) was substantial (0.40) (see their Table 1). They wrote “the big everything factor accounted for considerable variance in various domain indicators, with the exception of the performance-based measures”. Therefore, concluding that ‘cognitive ability is unrelated with personality or psychopathology’ might be premature.

In this later regard, the general factor of personality (P) was defined in the [35]’s ([35]) meta-analysis as the “ability to regulate behavior in order to achieve social goals” (emphasis added). Furthermore, [7] ([7]) indicated that “high-p individuals experience difficulties in regulation/control when dealing with others, the environment, and the self” (emphasis added). It is tempting to jump to the logical (and theoretically meaningful) connection of the regulatory mental processes associated to P and p with cognition (aka, intelligence) ([32]). If we as scientists are experiencing problems in uncovering this underlying theoretically reasonable connection, we should think of designing one proper playground. We visualize two ways out: (1) we administer self-reports for measuring all the cognitive and non-cognitive variables of interest or (2) we design and administer performance-based measures of these variables.

Sadly, the second approach seems especially complicated. Thus, for instance, [36] ([36]) designed GBAs (game-based assessments) for measuring three conscientiousness facets (achievement striving, self-discipline, and cautiousness). Regrettably, they did not find any relationship between performance on the GBAs and measures of self-reported personality. However, they found remarkable correlations between GBAs performance and cognitive ability (*r* = 0.50). They concluded that “future research developing personality-oriented GBAs should consider the contamination of personality by cognitive ability”. However, comprehensive previous research, summarized by [29] ([29]), shows that this ‘contamination’ is hardly avoidable. Moreover, researchers should be more inclined to assume that “personality affects performance on cognitive tests and cognitive ability affects item responses on personality assessments” ([21]). Modeling instead of controlling for their interplay may be much more interesting.

### 4.6. Limitations

The first obvious limitation is that we were unable to test what happens when cognitive and non-cognitive variables are assessed using self-reports or performance-based measures. As discussed, this method factor seems crucial to uncover the true relationships between these psychological factors.

The second limitation is that having a criterion measure is desirable to know how the considered cognitive and non-cognitive variables relate to everyday life behavior. [25]’s ([25]) study described above is an example. [34]’s ([34]) research is another. We were able to recover the scores obtained by our participants in the academic grades achieved before making their debut at the university. This allowed for the correlation values between these scores and the factors and variables considered here to be computed. The cognitive factor showed a positive and statistically significant correlation (*r* = 0.30, *p* < .01). Among the personality and psychopathological variables, only conscientiousness showed a significant correlation value (*r* = 0.15, *p* < .05). Appendix C shows further information regarding these academic grades.

Finally, when psychopathology is considered along with cognitive ability and personality traits, it may be illustrative to test clinical samples in addition to people from the general population without remarkable psychopathology symptoms. [22]’s ([22]) research examining high-risk college students (family history positive of alcoholism) is an example and [1] ([1]) stressed the issue that the ‘C factor’ is especially relevant for any form of psychopathology compared with non-psychiatric controls. Nevertheless, it is important to keep in mind that qualitative differences between normal and abnormal bands are not expected. We are studying dimensions not categories ([19]; [27]).

### 4.7. What to Conclude

Here, we have reported large, moderate, and small relationships among cognitive ability, personality traits, and subjective psychopathological symptoms. However, cognitive variables were performance based, whereas non-cognitive variables were based on self-report. This measurement distinction seems crucial for the tough scenario scientists encounter when addressing this three-way relationship.

Going one step further, it can be suggested that the sharp distinction between cognitive and non-cognitive variables might not be properly justified from a theoretical perspective. The key term ‘control’ appears in both the personality and psychopathological domains and this ‘control’ can be thought of as an unbeatable cognitive factor. In this regard, the underappreciated but comprehensive Systems Net Theory (SNT) proposed by [23] ([23]) integrates intelligence, personality, and psychopathology. He wrote that “self-control [conscientiousness] regulates temperamental dimensions [extraversion, agreeableness, and neuroticism] through information supplied by the intellect. The intellect assesses the fit between the functioning of the three temperamental dimensions and external demands. The joint end of these two dimensions is to self-regulate the three temperamental dimensions to enable the adjustment of behavior” (p. 220, emphasis added).

SNT perspective seems closely like “the intelligence-as-cause perspective” discussed by [3] ([3]). According to their rational analysis, intelligence (a) facilitates successfully completing cognitively demanding tasks, (b) drives academic, life, and career interests and choices, (c) increases willingness to enjoy tough mental and vital challenges, (d) leads to the avoidance of structure and routine, and (e) helps to navigate everyday life with smaller levels of stress.

Finally, it must be underscored that the key mark of the intellect is its integrative nature. Research has identified a myriad of mental abilities, but all must be integrated and orchestrated in some way to guide human behavior ([17]). We cannot see any powerful reason to exclude non-cognitive variables (personality and psychopathology) from the influence of this integrative general cognitive ability. [5] ([5]) noted that “all mental disorders are expressed through dysfunction of the same organ (the brain), whereas physical diseases such as cirrhosis, emphysema, and diabetes are manifested through dysfunction of different organ systems”. And as highlighted by [8] ([8]) “maybe there is no place in the brain for general intelligence, because the brain itself is the place. And we only have a single brain”. Change ‘general intelligence’ by ‘psychological traits’ and you will get the idea.

## Figures and Tables

**Table 1 jintelligence-12-00096-t001:** Correlation matrix.

	1	2	3	4	5	6	7	8	9	10	11	12	13	14	15	16	17
1. Gf																	
2. Gc	0.34 **																
3. Gv	0.40 **	0.27 **															
4. Gq	0.45 **	0.30 **	0.31 **														
5. Extraversion	−0.06	−0.03	−0.07	0.00													
6. Agreeableness	−0.11	−0.07	−0.12 *	−0.14 *	0.25 **												
7. Conscientiousness	−0.14 *	−0.12 *	−0.10	−0.17 **	0.12 *	0.10											
8. Neuroticism	0.02	−0.11	0.00	0.04	−0.28 **	−0.17 **	−0.25 **										
9. Openness	0.09	0.31 **	0.06	−0.07	0.05	0.13 *	0.00	0.05									
10. Somatization	−0.04	−0.05	−0.13 *	−0.03	−0.24 **	−0.14 *	−0.19 **	0.47 **	0.07								
11. Obsessive-Compulsive	−0.02	0.00	−0.13 *	0.00	−0.30 **	−0.07	−0.30 **	0.52 **	0.09	0.63 **							
12. Interpersonal sensitivity	0.05	0.04	−0.08	0.07	−0.42 **	−0.17 **	−0.21 **	0.58 **	0.07	0.56 **	0.67 **						
13. Depression	0.10	0.05	−0.04	0.09	−0.32 **	−0.13 *	−0.19 **	0.65 **	0.09	0.70 **	0.69 **	0.75 **					
14. Anxiety	−0.08	−0.05	−0.17 **	−0.01	−0.25 **	−0.13 *	−0.16 **	0.57 **	0.10	0.77 **	0.68 **	0.68 **	0.76 **				
15. Hostility	0.05	0.05	−0.01	0.14 *	−0.22 **	−0.33 **	−0.20 **	0.45 **	0.03	0.50 **	0.49 **	0.54 **	0.55 **	0.51 **			
16. Phobic anxiety	−0.11	−0.01	−0.09	−0.14 *	−0.31 **	−0.10	−0.17 **	0.38 **	0.00	0.65 **	0.56 **	0.63 **	0.58 **	0.65 **	0.41 **		
17. Paranoid ideation	0.04	0.00	−0.07	0.04	−0.22 **	−0.31 **	−0.10	0.38 **	0.03	0.52 **	0.60 **	0.72 **	0.60 **	0.58 **	0.54 **	0.56 **	
18. Psychoticism	0.00	0.03	−0.12 *	0.05	−0.31 **	−0.09	−0.19 **	0.49 **	0.11	0.63 **	0.68 **	0.74 **	0.72 **	0.72 **	0.54 **	0.60 **	0.68 **

Cognitive ability → Gf = reasoning/fluid ability, Gc = crystallized ability, Gv = visuospatial ability, Gq = quantitative ability; Personality (NEO-FFI) → extraversion, agreeableness, conscientiousness, neuroticism, openness; psychopathology (SCL-90-R) → somatization, obsessive, interpersonal sensitivity, depression, anxiety, hostility, phobias, paranoid ideas, psychoticism. * *p* < 0.05, ** *p* < 0.01.

**Table 2 jintelligence-12-00096-t002:** Principal axis factoring (PAF, eigenvalues greater than 1) followed by Promax rotation.

	F1 (p)	F2 (*g*)	F3 (Agreeableness)	F4 (Introversion)
Cognitive ability				
Gf	0.07	**0.69**	−0.17	0.07
Gc	0.02	**0.56**	0.18	0.03
Gv	−0.07	**0.49**	−0.05	0.09
Gq	0.08	**0.60**	−0.22	−0.04
Personality Traits				
Extraversion	−0.31	−0.08	0.15	**−0.53**
Agreeableness	−0.20	−0.19	**0.57**	−0.11
Conscientiousness	−0.29	−0.15	0.06	−0.19
Neuroticism	**0.63**	0.05	−0.18	0.25
Openness	0.11	0.19	0.29	−0.04
Subjective Psychopathological Symptoms				
Somatization	**0.78**	−0.04	−0.08	0.12
Obsessive	**0.78**	0.02	−0.07	0.34
Interpersonal	**0.82**	0.13	−0.21	**0.62**
Depression	**0.84**	0.18	−0.15	0.31
Anxiety	**0.85**	−0.05	−0.08	0.13
Hostility	**0.62**	0.22	**−0.45**	0.13
Phobias	**0.67**	−0.11	0.00	**0.40**
Paranoid	**0.69**	0.12	**−0.38**	0.32
Psychoticism	**0.80**	0.10	−0.09	**0.40**

## Data Availability

The dataset analyzed here might be available upon request from interested researchers.

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
