# Peer review of "Cognitive Ability, Personality, and Psychopathology: A Stormy Relationship"

_jintelligence, 2024, doi:10.3390/jintelligence12100096_

Round 1

Reviewer 1 Report

Comments and Suggestions for Authors

This is an interesting manuscript (ms.), and it has the potential for making a contribution to the literature.  That said, there are a few things that need to be addressed prior to a positive publication recommendation, as follows:

1.  The author(s) report that ‘personality facets relate to intelligence better r = .40 . . .  It is not clear that this is an ‘average’ or ‘mean’ correlation.  My recollection is that this is not the case from the meta-analysis they cite.  If that is the case, it is a substantial overestimate of the relationships between personality facets and intelligence.

2.  The use of ‘factor scores’ to represent the relations between factors and other variables is generally not recommended (e.g., see Tucker, L. R, 1971, Relations of factor score estimates to their use.  Psychometrika, 36(4), 427-436).  Instead, the author(s) should use something like the Dwyer Extension, which avoids the use of factor scores. (See, e.g., Gorsuch, R. L. 1997, New Procedure for extension analysis in exploratory factor analysis, 57(5), 725-740 for a review).

3.  The authors should report the “academic grades” variables in the various tables, not just in the “limitations” section (i.e., means, sds, correlations)

4.  Other than the academic grades variable(s), the authors need to take account of several considerations regarding the ability measures:
a.  There could be a restriction-of-range of talent, given that the sample was entirely composed of university students.
b.  As noted in the literature since Cronbach (1957), personality measures are aimed at assessing “typical” behavior, and ability tests are aimed at “maximal” performance.  Interpreting the correlations between these variables necessitates consideration that the testing ‘situation’ itself may be partially responsible for the shared variance among ability measures and personality traits.  That is, these relationships may over-estimate the relations at the trait/facet level in a way that might not be present for non-maximal performance measures of ability (e.g., ‘typical’ behaviors that require cognitive abilities).

Author Response

Reviewer 1 (R1)

We thank R1 for the general positive feedback.

1.- The author(s) report that ‘personality facets relate to intelligence better r = .40 . . .  It is not clear that this is an ‘average’ or ‘mean’ correlation.  My recollection is that this is not the case from the meta-analysis they cite.  If that is the case, it is a substantial overestimate of the relationships between personality facets and intelligence.

The reported values are adjusted multiple Rs (please see page 316 from Anglim et al., 2022). This is now specified to avoid confusions.

2.- The use of ‘factor scores’ to represent the relations between factors and other variables is generally not recommended (e.g., see Tucker, L. R, 1971, Relations of factor score estimates to their use.  Psychometrika, 36(4), 427-436).  Instead, the author(s) should use something like the Dwyer Extension, which avoids the use of factor scores. (See, e.g., Gorsuch, R. L. 1997, New Procedure for extension analysis in exploratory factor analysis, 57(5), 725-740 for a review).

It is unclear why the use of factor scores is not recommended given the main goal of our research, namely, to characterize the raw simultaneous relationships between cognitive and non-cognitive variables. From this perspective, we document/compute how these variables spontaneously relate in this dataset. Thus, for instance, Table 2 shows factor loadings on the four factors of the variables included in the dataset and the corresponding correlation matrix (Table 1). Factor scores are computed to illustrate the key argument unfolded across the Ms. The rotated factor solution shows how the variables are indeed related to different degrees. Anyways, the implications we extract from this solution are tempered and tentative for raising the key conceptual and methodological points (please refer to the discussion section).

3.- The authors should report the “academic grades” variables in the various tables, not just in the “limitations” section (i.e., means, sds, correlations)

The reporting of academic grades is secondary for this research. Therefore, we just use them just to document one minor, albeit interesting, point within the discussion section. Nevertheless, we now include the requested information in Appendix C.

4.- Other than the academic grades variable(s), the authors need to take account of several considerations regarding the ability measures:
a.  There could be a restriction-of-range of talent, given that the sample was entirely composed of university students.
b.  As noted in the literature since Cronbach (1957), personality measures are aimed at assessing “typical” behavior, and ability tests are aimed at “maximal” performance.  Interpreting the correlations between these variables necessitates consideration that the testing ‘situation’ itself may be partially responsible for the shared variance among ability measures and personality traits.  That is, these relationships may over-estimate the relations at the trait/facet level in a way that might not be present for non-maximal performance measures of ability (e.g., ‘typical’ behaviors that require cognitive abilities).

Regarding the ability measures (a) it is unclear if there is a remarkable restriction of talent these days; there is research raising serious doubts in this regard (Meta-analysis: on average, undergraduate students’ intelligence is merely average); and (b) the distinction maximal-typical is certainly interesting and it might contribute to separate the cognitive and non-cognitive worlds. However, this distinction departs from the core question we address in our research. Of course, cognitive and non-cognitive variables are distinguishable, but the (reasonable) default hypothesis is that they interact, especially if measured through comparable formats (which is key for our general perspective detailed in the discussion section). Moreover, evidence regarding the interplay between these cognitive and non-cognitive variables, and what this evidence suggests from a theoretical perspective, is detailed in the Ms.

We thank R1 for the comments and suggestions.

Reviewer 2 Report

Comments and Suggestions for Authors

The current study presents correlations and factor analyses of cognitive ability, Big Five personality, and psychopathology symptoms in 299 students.

There were several major strengths of this study. First, the sample size quite good given the scale of the data collection. Second, The comprehensive battery of ability assessments is excellent and the use of proctored administration is also a major strength.  Third, the other measures are also strong measures. Fourth, the inclusion of the psychopathology symptom measure adds a novel angle. 

The paper is also generally well written.

I make several suggestions for refining the manuscript below. Most importantly, I would remove (or substantially reduce the weighting) on discussion of correlations between latent factors extracted from factor analyses. I discuss reasons for this below. Instead, I think it would be much more valid and interesting to score general factors from the three classes of measures in isolation: i.e., a g factor from the ability measures; a p-factor from the psychopathology measures; and GFP from the Big Five. I suggest including those factors in your main correlation matrix. At the absolute least, I hope you include these variables in your correlation matrix.

ASSORTED POINTS

1. Regarding " the SCL-90-R for evaluating a range of subjective symptoms". Perhaps you could be more specific about what the SCL-90-R measures: "e.g., "symptoms of psychopathology". You might even want to write "Symptom Checklist-90-R". In particular, I think that some of your readers might be less familiar with the measure if they are coming from a non-clinical background. I think the title could also clarify what symptoms are being assessed. The word symptoms could refer to any health problem. Perhaps "Symptoms of Psychopathology" or something like that. 

2. I also wonder whether "cognitive ability" is more appropriate than "cognition". when I hear "cognition" I think of cognitive processes rather than individual differences. Cognition is part of personality as well.

3. Perhaps personal opinion, but I'm not sure if "A Stormy Relationship" in the title adds value.

4. Minor personal preference, but can you write 299 rather than " two hundred and ninety-nine" in abstract. Make it easier for the  reader to see your sample size. I would also include age and gender in abstract and perhaps country of data collection.

5. When discussing the findings of Stanek and Ones, it is important to note what corrections were applied. In particular, I think they corrected for self-other agreement for personality traits, and reliability for cognitive ability assessment. The self-other correction in particular increases observed correlation quite a bit. While it's fine to make such corrections, it's important to note what is being spoken, because obviously observed correlations will be much smaller when making such large corrections.

6. Regarding cognitive ability and GFP, you may also want to contrast the Stanek estimate of 0.36 with that obtained by Anglim (I think something around .06 from memory). Anglim used actual a large number of actual datasets (I think over 70 from memory) to derive a GFP and correlate with cognitive ability. Anyway, these are very different estimates, and it is worth considering which you find more valid. Personally, I find the lower estimate more compelling as this is more consistent with estimates of the correlation between neuroticism and intelligence, perhaps offset by the fact that traits like extraversion and conscientiousness are relatively uncorrelated with intelligence, whereas openness (the trait most related to intelligence) is somewhat less involved in GFP in some cases.

7. Table 1 suggestions: Make the correlations lower diagonal (it's easier to read); remove the leading zero. Put  fulll names in first column with 1., 2., etc. and put numers in columns. I would order Big Five as NEOAC (unless you have a specific reason for your current ordering). Don't write "0" for a correlation, write ".00" 

8. I think you should also score the measures for general intelligence (e.g., a simple equal weighted sum of all measures; or the first unrotated principal component, or some other principled scoring), and include that in the correlation matrix. This would provide a big picture perspective from which the many subscales could be compared.

9. I would consider also including a GFP (either the first unrotated factor of the personality factors or items; or a simple weighted sum of the Big five after reversing N) and an overall psychopathology factor.

10. Regarding " The values for the cognitive variables ranged from 0.27 to 0.45". You could state this more clearly. You should say "The intercorrelations between cognitive ability..." And same for personality and symptoms.

11. Regarding "Four factors were obtained. " No. You chose four factors. You should specify how four factors were obtained. What rule did you apply?

12. I'm trying to work out the value of the four factor solution. It shows that psychopathology symptoms group together and that they are most related to neuroticism, although they also link in with some GFP type traits more weakly; cognitive ability forms a factor, which openness loading maximally. And then we start to pull out some of the other big Five factors which link in to specific psychopathology. I guess, we could ask what is missing. Presumably, a big five researcher would say that conscientiousness and openness is missing. But that said, it is hard to get a factor when there is only one variable represented.

13. Figure 1. I think this could be removed or relegated to a supplement. A correlation matrix would presumably suffice. I suppose there is a little skew with the p factor, but that ultimately is not that noteworthy.

14. I don't think the correlation between the factor scores map that well onto the literature. In particular, the "cognitive factor" has loadings from the non-cognitive ability items. And the personality and symptom factors also have loadings from the cognitive factors. I would find it more meaningful if you would (at least in addition) compute a general psychopathology factor from the symptoms measures, a GFP from the Big Five, and g factor from the cognitive tests. These would pure to their respective measures and frameworks. At the very least, I would really value it if you were to add these variables somewhere and report the correlations (ideally, I think they would go in the main correlation matrix).

15. Furthermore, factor correlations derived from factor analysis can be arbitrarily modulated by parameters of how the rotation is performed. So I really would not put as much weight on the correlations between these derived factor scores. As mentioned above, I think you'll be in a much stronger position if you derive actual general factors from respective measures.

16. You might also wonder consider what would happen if you corrected correlations. While I'm always wary of this, it would be interesting to see if you corrected correlations between cognitive ability and traits/psychopathology for self-other agreement prior to factor analysis. Specifically, correlation between objective and self-report correlations are substantially attenuated due to the different measurement approaches. For instance, self-reported intelligence and objective intelligence only correlate about .35 or something (see Freund). 

17. Regarding "For interpreting the correlation values..." I think given the attenuation observed when correlating self-report with objective ability measures, my own heuristic is that you have to roughly double such correlations if you are interested in relationships between latent constructs. For instance, the averaged observed correlations between g and openness are around .18 to .20, but I would consider these fairly large. Anyway, perhaps I'm expressing an opinion here, but hopefully you agree that in some sense our heuristics for effect sizes need to differ when considering self-report with self-report versus self-report with objectively assessed ability.

18. I wonder whether you have considered making the data available on the OSF.

Author Response

Reviewer 2 (R2)

We appreciate the general positive feedback regarding our Ms.

We tried to address in the revised version most of the points raised by R2 and we think the changes made improved the report.

1.- Regarding " the SCL-90-R for evaluating a range of subjective symptoms". Perhaps you could be more specific about what the SCL-90-R measures: "e.g., "symptoms of psychopathology". You might even want to write "Symptom Checklist-90-R". In particular, I think that some of your readers might be less familiar with the measure if they are coming from a non-clinical background. I think the title could also clarify what symptoms are being assessed. The word symptoms could refer to any health problem. Perhaps "Symptoms of Psychopathology" or something like that. 

Thanks for the suggestion. Now we use the terms ‘Symptoms of Psychopathology’ or ‘Psychopathological Symptoms’

2.- I also wonder whether "cognitive ability" is more appropriate than "cognition". when I hear "cognition" I think of cognitive processes rather than individual differences. Cognition is part of personality as well.

This is an interesting point. R2 states that “cognition is part of personality”. Well, this might be so, but also the other way around (a key conceptual argument of our report). We can say that personality is shaped by cognitive ability to some degree and, therefore, that the former can be considered as ‘charged’ by cognition.

Anyways, we now use the term ‘cognitive ability’ throughout the Ms., as suggested.

3.- Perhaps personal opinion, but I'm not sure if "A Stormy Relationship" in the title adds value.

We choose to keep the title to get the readers’ attention and because we think it describes the reiterative scientific questions regarding the relationship between cognitive and non-cognitive psychological variables, as detailed in the Ms.

4.- Minor personal preference, but can you write 299 rather than " two hundred and ninety-nine" in abstract. Make it easier for the  reader to see your sample size. I would also include age and gender in abstract and perhaps country of data collection.

Done. Thanks for the suggestion.

5.- When discussing the findings of Stanek and Ones, it is important to note what corrections were applied. In particular, I think they corrected for self-other agreement for personality traits, and reliability for cognitive ability assessment. The self-other correction in particular increases observed correlation quite a bit. While it's fine to make such corrections, it's important to note what is being spoken, because obviously observed correlations will be much smaller when making such large corrections.

Done. Now we specify that the relationships reported by Stanek and Ones are corrected.

6.- Regarding cognitive ability and GFP, you may also want to contrast the Stanek estimate of 0.36 with that obtained by Anglim (I think something around .06 from memory). Anglim used actual a large number of actual datasets (I think over 70 from memory) to derive a GFP and correlate with cognitive ability. Anyway, these are very different estimates, and it is worth considering which you find more valid. Personally, I find the lower estimate more compelling as this is more consistent with estimates of the correlation between neuroticism and intelligence, perhaps offset by the fact that traits like extraversion and conscientiousness are relatively uncorrelated with intelligence, whereas openness (the trait most related to intelligence) is somewhat less involved in GFP in some cases.

This is interesting but we think discussing this point is not especially relevant for our main purpose here. Discrepancies might come from the input data, methodological choices, etc. The important point is that they found theoretically meaningful relationships between cognitive and non-cognitive variables.

7.- Table 1 suggestions: Make the correlations lower diagonal (it's easier to read); remove the leading zero. Put  fulll names in first column with 1., 2., etc. and put numers in columns. I would order Big Five as NEOAC (unless you have a specific reason for your current ordering). Don't write "0" for a correlation, write ".00" 

Table 1 is now modified, as suggested. We did the same for the table shown in Appendix A.

8 & 9.- I think you should also score the measures for general intelligence (e.g., a simple equal weighted sum of all measures; or the first unrotated principal component, or some other principled scoring), and include that in the correlation matrix. This would provide a big picture perspective from which the many subscales could be compared.

I would consider also including a GFP (either the first unrotated factor of the personality factors or items; or a simple weighted sum of the Big five after reversing N) and an overall psychopathology factor.

As suggested, we computed separate factor scores for cognitive ability, personality, and psychopathological symptoms. We include this information in the revised version (please see Appendix B). But we do this for illustrative purposes only because this is not the way we choose to analyze this dataset (please see response 2 to R1 above).

10.- Regarding " The values for the cognitive variables ranged from 0.27 to 0.45". You could state this more clearly. You should say "The intercorrelations between cognitive ability..." And same for personality and symptoms.

Done. Thanks for the suggestion.

11.- Regarding "Four factors were obtained. " No. You chose four factors. You should specify how four factors were obtained. What rule did you apply?

This is correct. Thanks for noting. We deleted ‘Four factors were obtained’ from the revised version specifying the applied computation rule.

12.- I'm trying to work out the value of the four factor solution. It shows that psychopathology symptoms group together and that they are most related to neuroticism, although they also link in with some GFP type traits more weakly; cognitive ability forms a factor, which openness loading maximally. And then we start to pull out some of the other big Five factors which link in to specific psychopathology. I guess, we could ask what is missing. Presumably, a big five researcher would say that conscientiousness and openness is missing. But that said, it is hard to get a factor when there is only one variable represented.

We provide an interpretation of the factor solution, albeit it is not big deal for the main purpose of the report, as noted above. R2 writes: “it is hard to get a factor when there is only one variable represented”. We do not get what R2 implies with this sentence. The four factors look meaningful and they are loaded by at least three variables.

13.- Figure 1. I think this could be removed or relegated to a supplement. A correlation matrix would presumably suffice. I suppose there is a little skew with the p factor, but that ultimately is not that noteworthy.

Figure 1 is deleted, as suggested. R2 is right: the figure adds nothing.

14.- I don't think the correlation between the factor scores map that well onto the literature. In particular, the "cognitive factor" has loadings from the non-cognitive ability items. And the personality and symptom factors also have loadings from the cognitive factors. I would find it more meaningful if you would (at least in addition) compute a general psychopathology factor from the symptoms measures, a GFP from the Big Five, and g factor from the cognitive tests. These would pure to their respective measures and frameworks. At the very least, I would really value it if you were to add these variables somewhere and report the correlations (ideally, I think they would go in the main correlation matrix).

R2 writes: “I don't think the correlation between the factor scores map that well onto the literature. In particular, the "cognitive factor" has loadings from the non-cognitive ability items. And the personality and symptom factors also have loadings from the cognitive factors.”

This is exactly the key point of our report: how cognitive and non-cognitive factors are indeed related. This can be clearly seen in the factor matrix where loadings from different variables show up across factors. Nevertheless, as shown in Appendix B, we computed separate factor analyses for the three domains, as suggested (see above).

15.- Furthermore, factor correlations derived from factor analysis can be arbitrarily modulated by parameters of how the rotation is performed. So I really would not put as much weight on the correlations between these derived factor scores. As mentioned above, I think you'll be in a much stronger position if you derive actual general factors from respective measures.

Please note that we do not make a strong case from the correlations among factor scores. We just report the values making some suggestions afterwards. Importantly, because we want to summarize the relationships among variables, as shown in the correlation matrix, an oblique rotation was chosen to keep this crucial information.

16.- You might also wonder consider what would happen if you corrected correlations. While I'm always wary of this, it would be interesting to see if you corrected correlations between cognitive ability and traits/psychopathology for self-other agreement prior to factor analysis. Specifically, correlation between objective and self-report correlations are substantially attenuated due to the different measurement approaches. For instance, self-reported intelligence and objective intelligence only correlate about .35 or something (see Freund). 

This is an interesting suggestion, but we prefer to report raw instead of corrected correlations.

17.- Regarding "For interpreting the correlation values..." I think given the attenuation observed when correlating self-report with objective ability measures, my own heuristic is that you have to roughly double such correlations if you are interested in relationships between latent constructs. For instance, the averaged observed correlations between g and openness are around .18 to .20, but I would consider these fairly large. Anyway, perhaps I'm expressing an opinion here, but hopefully you agree that in some sense our heuristics for effect sizes need to differ when considering self-report with self-report versus self-report with objectively assessed ability.

R2 writes: “hopefully you agree that in some sense our heuristics for effect sizes need to differ when considering self-report with self-report versus self-report with objectively assessed ability.”

We agree but, again, we choose to report the raw correlations.

18.- I wonder whether you have considered making the data available on the OSF.

We are more than willing to share our dataset upon request. We acknowledge that there are other ways for looking at this dataset and we are open to share and learn from other approaches.

We thank R2 for the exhaustive analysis of our report. We strongly think the changes made after R2 comments and suggestions improved the Ms.

Round 2

Reviewer 1 Report

Comments and Suggestions for Authors

While I don't agree with some of the responses made by the author(s), I think that the author(s) have made an attempt to deal with these issues in a reasonable manner.  The paper could be improved with better attention to the stated concerns (e.g., factor scores), but at this point, it appears the authors are not able/willing to do so.  I think the paper could be improved significantly, but 'as is' it is publishable.

Author Response

We thank reviewer 1 for saying that, even when there are some disagreements, the paper is publishable 'as is'.”

Reviewer 2 Report

Comments and Suggestions for Authors

I was reviewer 2 on the initial submission. In general, I found their response to my suggestions to be pretty good and generally attentive. I still think it's important to be when in interpreting the overlap of GFP, p, and g to not rely too much on the factor analysis. I also think that care is needed when considering the meaning of the factor analysis of scales given that several key dimensions of individual differences, particularly in the Big Five domain are represented by a single scale.

REVIEW OF ROUND 1 COMMETS

1.     okay.

2.     okay.

3.     Okay. Just a difference of opinion.

4.     Good.

5.     Good.

6.     I disagree. I think the Anglim et al meta-analysis suggests that there is nothing special about the GFP correlation with intelligence. Rather it is consistent with neuroticism being related to intelligence and the GFP largely reflecting some of this contribution. In contrast, the Stanek et al estimate suggests that there is something special about GFP. Anyway, I agree it's not a deal breaker, but I think you should think about it. I think there are a lot more artefacts that could arise from the Stanek et al study.

7.     Great.

8.     Okay.

9.     Good.

10.  Good.

11.  good.

12.  Regarding " it is hard to get a factor when there is only one variable represented”. We do not get what R2 implies with this sentence." My point is that there are questions about the purpose of the factor analysis. If the purpose is to provide a sense of the structure of psychopathology, normal personality and intelligence, then I think an intuition would be that Big five plus cognitive ability would provide a reasonable representation, but if a domain is represented by only one variable, then factor analysis will not able to detect this very well. Generally, you need about 3 or more items to represent a factor. Anyway, this is something to think about when interpreting your factor analysis.

13.  Great.

14.  Thank you.

15.  I'm not sure if I really agree. I think this is a pretty key point. You are trying to understand the relationship between GFP, p, and g. That's the point of the paper. Anyway, I won't stress the point. But I think this is something to be very careful with in your interpretation.

16.  I also prefer the raw correlations and agree that should be the focus. This was more of a supplementary analysis suggestion.

17.  okay

18.  I understand. It's disappointing. In general, I think that sharing the data is a nice way to contribute to science and reassure readers about replicability of results. The authors identified no ethical or other reason why they could not share the data.

·       

 [JA1]The current study presents correlations and factor analyses of cognitive ability, Big Five personality, and psychopathology symptoms in 299 students.

There were several major strengths of this study. First, the sample size quite good given the scale of the data collection. Second, The comprehensive battery of ability assessments is excellent and the use of proctored administration is also a major strength.  Third, the other measures are also strong measures. Fourth, the inclusion of the psychopathology symptom measure adds a novel angle.

The paper is also generally well written.

I make several suggestions for refining the manuscript below.

Most importantly, I would remove (or substantially reduce the weighting) on discussion of correlations between latent factors extracted from factor analyses. I discuss reasons for this below. Instead, I think it would be much more valid and interesting to score general factors from the three classes of measures in isolation: i.e., a g factor from the ability measures; a p-factor from the psychopathology measures; and GFP from the Big Five.

I suggest including those factors in your main correlation matrix. At the absolute least, I hope you include these variables in your correlation matrix.

Author Response

We thank reviewer 2 for acknowledging that our “responses to the suggestions are pretty good and generally attentive.” Nevertheless, R2 thinks we must be careful when interpreting the results of the computed factor analysis. R2 is right, albeit we believe we are already careful, as noted in the first round of reviews:

“We document/compute how these variables spontaneously relate in this dataset. Thus, for instance, Table 2 shows factor loadings on the four factors of the variables included in the dataset and the corresponding correlation matrix (Table 1). Factor scores are computed to illustrate the key argument unfolded across the Ms. The rotated factor solution shows how the variables are indeed related to different degrees. Anyways, the implications we extract from this solution are tempered and tentative for raising the key conceptual and methodological points (please refer to the discussion section).

We provide an interpretation of the factor solution, albeit it is not big deal for the main purpose of the report.”